# Exosome component 1 cleaves single-stranded DNA and sensitizes human kidney renal clear cell carcinoma cells to poly(ADP-ribose) polymerase inhibitor

Qiaoling Liu[1†], Qi Xiao[1†], Zhen Sun[1†], Bo Wang[2†], Lina Wang[1], Na Wang[1], Kai Wang[1], Chengli Song[1*], Qingkai Yang[1*]

[1]Institute of Cancer Stem Cell, DaLian Medical University, Dalian, China; [2]Department of General Surgery, Second Affiliated Hospital, DaLian Medical University, Dalian, China

**Abstract** Targeting DNA repair pathway offers an important therapeutic strategy for *Homo sapiens* (human) cancers. However, the failure of DNA repair inhibitors to markedly benefit patients necessitates the development of new strategies. Here, we show that exosome component 1 (EXOSC1) promotes DNA damages and sensitizes human kidney renal clear cell carcinoma (KIRC) cells to DNA repair inhibitor. Considering that endogenous source of mutation (ESM) constantly assaults genomic DNA and likely sensitizes human cancer cells to the inhibitor, we first analyzed the statistical relationship between the expression of individual genes and the mutations for KIRC. Among the candidates, EXOSC1 most notably promoted DNA damages and subsequent mutations via preferentially cleaving C site(s) in single-stranded DNA. Consistently, EXOSC1 was more significantly correlated with C>A transversions in coding strands than these in template strands in human KIRC. Notably, KIRC patients with high EXOSC1 showed a poor prognosis, and EXOSC1 sensitized human cancer cells to poly(ADP-ribose) polymerase inhibitors. These results show that EXOSC1 acts as an ESM in KIRC, and targeting EXOSC1 might be a potential therapeutic strategy.

*For correspondence:
chenglisong2015@hotmail.com (CS);
yangqingkai@dmu.edu.cn (QY)

[†]These authors contributed equally to this work

Competing interests: The authors declare that no competing interests exist.

## Introduction

DNA damages and subsequent mutations are central to development, progression, and treatment of nearly all cancers (*Brown et al., 2017*; *Farmer et al., 2005*; *Jeggo et al., 2016*; *Pearl et al., 2015*; *Roos et al., 2016*). Cancer cells frequently decrease DNA repair pathways and increase endogenous sources of mutation (ESM) to drive mutations (*Brown et al., 2017*; *Farmer et al., 2005*; *Jeggo et al., 2016*; *Pearl et al., 2015*; *Roos et al., 2016*). Hence, cancer cells are often more reliant on a subset of DNA repair pathway(s) to survive DNA damages. Targeting critical DNA repair members, such as poly (ADP-ribose) polymerases (PARPs) (*Brown et al., 2017*; *Tubbs and Nussenzweig, 2017*), offers a therapeutic strategy for cancers (*Tutt et al., 2010*). Inhibition of PARPs by small-molecule compounds disrupts the ability of cancer cells to survive ongoing DNA damage and results in cell cycle arrest and/or cell death (*Lord and Ashworth, 2012*). However, the failure of PARP inhibitors (PARPis) to markedly benefit patients suggests the necessity for developing new strategies. Due to the central role of ESM in ongoing DNA damages, there is a need for the identification and understanding of ESM.

ESM constantly assaults genomic DNA and almost inevitably leads to mutations (*Figure 1A*; *Jeggo et al., 2016*; *Roos et al., 2016*). However, most of the ESM studies were focused on deamination. The significance of deamination as an ESM is supported mainly by two observations: (1) Transitions show higher frequency than expected by chance, although there are twice as many possible

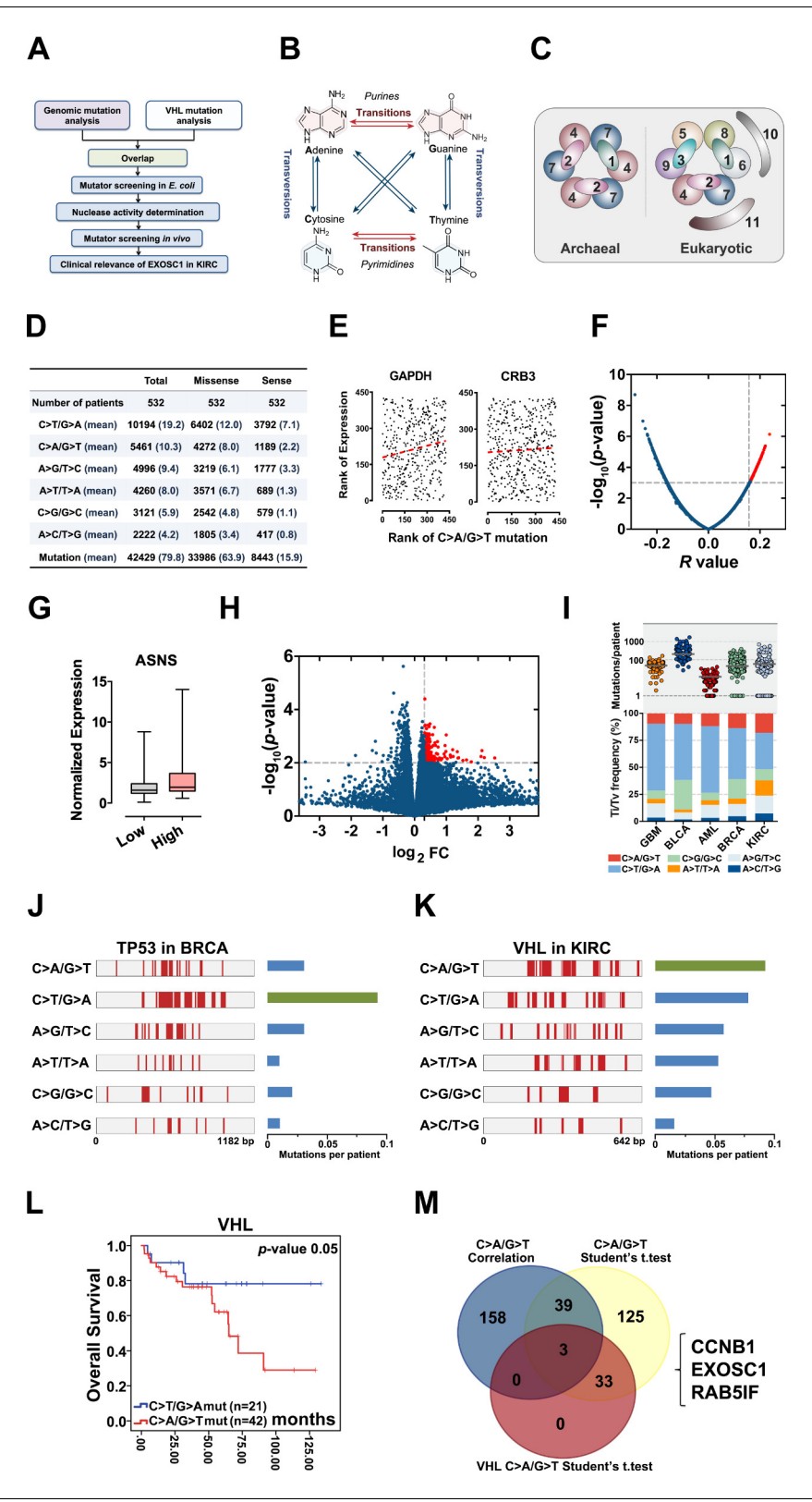

**Figure 1.** Identification of candidate ESMs in KIRC by statistical analyses. (**A**) Schematic of this study. (**B**) Illustration of base substitutions. (**C**) Schematic showing the archaeal and eukaryotic exosome complexes viewed from the top. (**D**) Summary statistics for the six types of c-substitutions in KIRC. (**E**) Scatter plots showing the correlation between the rank of mutation and gene expression. Each plot represents one KIRC sample. The orange dashed line shows the best fit for visualization. P values were calculated by Spearman's rank correlation. (**F**) Volcano plots of p and r values calculated by

*Figure 1 continued on next page*

*Figure 1 continued*

Spearman's correlation analyses. Each plot represents one gene. The top 1% of genes were taken as candidates and marked in red. (**G**) Box plots showing ASNS expression in the high and low C>A/G>T mutation groups. The expression was normalized to TBP. (**H**) Volcano plots showing the p and fold change (FC) values calculated by the two-tailed Student's *t*-test. Each plot represents one gene. FC was calculated by the formula: FC=the mean gene expression in the high group/that in the low group. The top 1% of genes were taken as candidates and marked in red. (**I**) C-substitution mutation frequencies in five types of major cancers. (**J, K**) Mutation spectra of the TP53 gene in BRCA (**J**) and VHL gene in KIRC (**L**). (**L**) Kaplan-Meier (KM) analyses of OS between VHL C>A/G>T and C>T/G>A mutation groups. The median OSs in the C>A/G>T and C>T/G>A groups were 72.95 and 108.91 months, respectively. The p value was obtained from the log-rank test. (**M**) Venn diagram showing the overlap of the candidate genes identified by three types of statistical analyses as noted. ESM, endogenous source of mutation; KIRC, kidney renal clear cell carcinoma; OS, overall survival. The online version of this article includes the following source data and figure supplement(s) for figure 1:

**Source data 1.** Identification of candidate ESMs in KIRC by statistical analyses.

**Figure supplement 1.** Identification of candidate ESMs in KIRC by statistical analyses.

**Figure supplement 1—source data 1.** Identification of candidate ESMs in KIRC by statistical analyses.

transversions. Nucleotide substitutions consist of two types: transition and transversion (*Alexandrov et al., 2013*; *Burgess, 2019*; *Petljak et al., 2019*). Transition is a substitution in which one base is replaced by another of the same class (purine or pyrimidine), while transversion is a substitution in which a purine is replaced with a pyrimidine or vice versa (*Figure 1B*). (2) C>T transitions at methylated cytosine in CG base pairs display a higher frequency than expected (*Alexandrov et al., 2013*; *Alexandrov et al., 2013*; *Burgess, 2019*; *Hutchinson, 2013*; *Petljak et al., 2019*). Therefore, activation-induced cytidine deaminase (AID) (*Barlow et al., 2013*; *Greaves, 2018*; *Petersen-Mahrt et al., 2002*) and apolipoprotein B mRNA editing enzyme catalytic (APOBEC) family (*Alexandrov et al., 2013*; *Buisson et al., 2019*; *Hutchinson, 2013*; *McGranahan et al., 2017*; *Robertson et al., 2017*) catalyzing the deamination of C were identified as ESMs. Unfortunately, some cancers, such as kidney renal clear cell carcinoma (KIRC), show the low mutation proportion at CG base pairs and low APOBEC expression (*Burns et al., 2013b*), raising the potential roles of unidentified ESMs.

Due to advances in sequencing technology and the great efforts of The Cancer Genome Atlas (TCGA), it is now possible to explore the statistical relationship between mutations and the expression of individual genes in multiple cancer types (*Weinstein et al., 2013*). The majority of patients included in the TCGA database are accompanied by data regarding both mutations and genome-wide expression of individual genes (*Weinstein et al., 2013*). Because that DNA damages often comprise a major source of mutation, the relativity between DNA damage and mutation allows quantitative analyses of mutation to be taken as a proxy of DNA damage (*Brown et al., 2017*; *Jeggo et al., 2016*; *Pearl et al., 2015*). Furthermore, the gene-specific correlation between mRNA and protein levels allows quantitative analyses of individual gene expression as an indicator for the corresponding protein (*Peng et al., 2015*; *Uhlen et al., 2017*; *Zhang et al., 2017*). Hence, analyses of the cancer cohort may identify candidate ESMs (*Tubbs and Nussenzweig, 2017*).

The exosome is an evolutionarily conserved multiprotein complex formed by exosome components (EXOSCs) (*Bousquet-Antonelli et al., 2000*; *Brown et al., 2000*; *Tomecki et al., 2010*). In eukaryotes, the exosome complex has a 'ring complex' (EXOSC4–EXOSC9) and a 'cap' structure (EXOSC1–EXOSC3) (*Figure 1C*). The human exosome complex may also contain two additional subunits, EXOSC10 and EXOSC11 (*Houseley et al., 2006*; *Lorentzen et al., 2008*; *Liu et al., 2006*; *Wasmuth et al., 2014*), which provide 3′–5′ exo- and/or endoribonuclease activities (*Januszyk and Lima, 2014*; *Kilchert et al., 2016*). The exosome is well known to degrade RNA (*Januszyk and Lima, 2014*; *Kilchert et al., 2016*; *Ulmke et al., 2021*). Hence, the exosome was reported to protect cells from genomic instability via degrading the DNA/RNA hybrids and restricting DNA strand mutational asymmetry (*Lim et al., 2017*; *Pefanis and Basu, 2015*; *Pefanis et al., 2015*). Interestingly, the cap unit EXOSC2 is stably associated with the exosome complex, while EXOSC1 is not stably associated (*Dai et al., 2018*; *Malet et al., 2010*), suggesting that EXOSC1 might be involved in some functions independent of the complex.

In this study, we show that EXOSC1 acts as an ESM and sensitizes cancer cells to PARPi in KIRC. Due to the role of exosome in maintaining genomic stability, these results also indicate that a unit of multiprotein complex can play a role opposite to that of the complex.

# Results

## Identification of candidate ESMs in KIRC

Because that ESMs constantly assault genomic DNA, we hypothesized that ESMs likely sensitized cancer cells to the inhibitors of DNA repair pathways. Considering that substitution is the most abundant mutation in all cancers, we initiated this study to identify the candidate ESMs responsible for substitution mutations. To identify the candidate ESMs other than deamination, we focused on KIRC for three reasons: (1) KIRC shows the lowest proportion of mutations at CG in major cancer types (*Burns et al., 2013b*), suggesting that the deamination contributes less to the mutations in KIRC. (2) Only low expressions of AID and APOBECs were detected in KIRC (*Burns et al., 2013b*). (3) The kidney potentially suffers less from exogenous source of mutations (EOSMs) (*Loeb, 2011*; *Roberts and Gordenin, 2014*). RNA-seq and exomic mutation data corresponding to 532 KIRC patients and 30,254 somatic substitution mutations in the TCGA were retrieved from The cBio Cancer Genomics Portal (http://cbioportal.org) (*Figure 1D*). Because that ESMs likely show a similar impact on the template and code DNA strands, the 12 types of substitution were groups into six types of complementary substitution (c-substitution) to simplify the analyses (*Figure 1D*).

Spearman's rank analysis was first performed to assess the correlation between each c-substitution type and the genome-wide expression of individual genes. Resultant p and r values were used for the further analyses. For example, GAPDH showed a p=0.0011 and r=0.16 correlation with C>A/G>T c-substitution, indicating that GAPDH expression displayed a positive correlation with C>A/G>T (*Figure 1E* and *Figure 1—source data 1*). Similarly, CRB3 was also positively correlated with C>A/G>T (*Figure 1E* and *Figure 1—source data 1*). Although the p values of multiple genes were lower than 0.05 (*Figure 1F* and *Figure 1—source data 1*), only top-ranked 200 genes (approximately 1% of the genome-wide genes) were taken as the candidates for each c-substitution type (*Supplementary file 1*). Gene Ontology (GO) enrichment analyses indicated that, generally, these top candidate genes showed the enrichment in 'mitochondrial gene expression' and 'organophosphate biosynthetic process' (*Figure 1—figure supplement 1A* and *Figure 1—figure supplement 1—source data 1*).

Student's *t*-test analysis was then used to determine whether the expression difference of individual genes between the high and low c-substitution groups is significant. The expression of individual genes in each patient was normalized by a housekeeping gene, TATA-binding protein (TBP), as previously described (*Burns et al., 2013a*; *Burns et al., 2013b*). According to each c-substitution, 532 KIRC patients were grouped into three groups (high, medium, and low). The difference of individual genes between the high and low c-substitution mutation groups was then analyzed by Student's *t*-test. Resultant p and fold change (FC) values were used for further analyses. For example, ASNS with p=0.0005 and FC=1.39 indicates that ASNS was increased in the high group (*Figure 1G* and *Figure 1—source data 1*). Although the p values of multiple genes were lower than 0.05 (*Figure 1H* and *Figure 1—source data 1*), only the top 200 genes with high FC and p<0.05 were taken as the candidates (*Supplementary file 2*). Notably, none of the APOBEC family members were identified as candidate by correlation or Student's *t*-test analyses, supporting that deamination contributes less to the mutations in KIRC.

Next, we performed meta-analyses to determine which of the six c-substitution types to focus on. Mutation frequencies of c-substitution types were first analyzed in five major cancers: breast adenocarcinoma (BRCA), glioblastoma multiforme (GBM), bladder urothelial carcinoma (BLCA), acute myeloid leukemia (AML), and KIRC, which potentially suffer less from the EOSMs. Although most frequently mutated substitutions in the five major cancers were C>T/G>A, KIRC displayed higher frequencies of C>A/G>T, A>T/T>A, and A>C/ >G mutations than the other four cancers did (*Figure 1I* and *Figure 1—source data 1*). Among the KIRC patient mutations, most c-substitutions are function-related (*Figure 1—figure supplement 1B*). Notably, the transversions in KIRC showed more capacities to result in function-related mutations than transitions (C>T/G>A and A>G/T>C) did (*Figure 1—figure supplement 1B*). Using tumor protein p53 (TP53) as control, we then assessed the frequencies of c-substitution types in von Hippel-Lindau tumor suppressor (VHL), the most frequently mutated gene in KIRC. Consistent with previous studies (*Burns et al., 2013a*; *Kandoth et al., 2013*), the most frequent c-substitution type of TP53 mutations in BRCA was C>T/G>A (*Figure 1J* and *Figure 1—source data 1*), while the most frequent type of VHL mutations in KIRC was C>A/G>T (p=0.004, chi-squared test) (*Figure 1K* and *Figure 1—source data 1*). Even after normalization

according to the base frequency, this phenomenon was still observed (p=0.021) (*Figure 1—figure supplement 1C and D*, and *Figure 1—figure supplement 1—source data 1*). Further, Kaplan–Meier (KM) analysis of overall survival (OS) indicated that patients with VHL C>A/G>T mutations showed poor OS (*Figure 1L*). These observations raised the significance of C>A/G>T c-substitutions in KIRC.

We then evaluated the expression difference of individual genes between the VHL C>A/G>T mutation-positive and mutation-negative patients. Student's *t*-tests analyses showed that 36 genes displayed p<0.05 (*Supplementary file 3*). Further overlap analyses demonstrated that cyclin B1 (CCNB1), exosome component 1 (EXOSC1), and RAB5 interacting factor (RAB5IF) were identified as candidate ESMs for C>A/G>T by all of the above analyses (*Figure 1M*).

## EXOSC1 promotes mutations in *Escherichia coli*

To evaluate the capability of the candidate gene to promote mutation, rifampicin-resistant assay in *Escherichia coli* was performed as previously described (*Petersen-Mahrt et al., 2002*). Because that mutation of the rifampicin-targeted rpoB gene to rifampicin resistance (Rif$^R$) occurs at a low frequency, the capability of a gene to mutate rpoB to Rif$^R$ can be evaluated by fluctuation analysis (*Petersen-Mahrt et al., 2002*). Therefore, AID, an known ESM (*Petersen-Mahrt et al., 2002*), was used as a positive control. Four genes (CDK5, TARBP2, PSAT1, and NECAB3) were used as random controls (*Figure 2—figure supplement 1A* and *Supplementary file 4*). These genes were expressed in *E. coli* under the regulation of a trp/lac (tac) hybrid promoter, which could be activated by isopropyl β-D-1-thiogalactopyranoside (IPTG) (*Figure 2A*). Consistent with a previous study (*Petersen-Mahrt et al., 2002*), AID enhanced mutation in *E. coli* (*Figure 2B* and *Figure 2—source data 1*). Notably, EXOSC1 more significantly increased mutations than AID did (p=4.08 × 10$^{-5}$) (*Figure 2B* and *Figure 2—source data 1*). We then evaluated the capabilities of EXOSC1 homologs (EXOSC2–EXOSC9) to promote mutations (*Figure 2—figure supplement 1B*). Among the members of exosome complex, EXOSC1 most notably enhanced mutation in *E. coli* (p=3.5 × 10$^{-11}$) (*Figure 2C* and *Figure 2—source data 1*). To determine whether the increase in mutation frequency stemmed from EXOSC1 protein itself, rifampicin-resistant assays were performed in the presence or absence of IPTG, the transcriptional inducer. As shown in *Figure 2E–F*, IPTG absence notably decreased the mutation frequency (p=1.29 × 10$^{-9}$), indicating that the protein of EXOSC1 promoted the mutation. Additionally, we evaluated the impact of EXOSC1 on the growth of *E. coli*. As shown in *Figure 2—figure supplement 1C*, EXOSC1 expression only slightly decreased cell growth, which might be due to the increase in mutation burden (*Schaaper and Dunn, 1987*).

Next, the mutation spectra of Rif$^R$ were analyzed by sequencing the rpoB gene PCR products from rifampicin-resistant clones. Sequencing of rpoB gene in 25 randomly selected rifampicin-resistant clones indicated that most of mutations in control clones were C>T/G>A transitions, while EXOSC1 frequently promoted C>A/G>T transversion mutations (*Figure 2G–I* and *Figure 2—source data 1*). Moreover, most of C>A/G>T transversions in EXOSC1-transformed cells were clustered at C1576 (10/36 mutations) and C1699 (6/36 mutations), whereas C>T/G>A transitions in control cells showed a distinct distribution with major hot spots at C1565 (6/32 mutations) and C1721 (5/32 mutations) (*Figure 2G* and *Figure 2—source data 1*). Hence, it was suggested that the mutations in control and EXOSC1-transformed cells were promoted by a distinct mechanism. Interestingly, EXOSC1-transformed cells also showed a shift of C>A/G>T mutations from 6% (2/32 mutations) to 69% (25/36 mutations) (p=4.03 × 10$^{-7}$, chi-squared test) (*Figure 2H–J*). Even after normalization to the base frequency, this phenomenon was still significant (p=1.47 × 10$^{-6}$) (*Figure 2—figure supplement 1D* and *Figure 2—figure supplement 1—source data 1*).

## EXOSC1 cleaves single-stranded DNA

Considering that the exosome is well known to degrade RNA, we speculated that EXOSC1 might promote mutation through cleaving DNA. Therefore, EXOSC1 was expressed and purified in vitro (*Figure 3A*). The resultant EXOSC1 protein was incubated with generic single-stranded DNA (ssDNA), double-stranded DNA (dsDNA), or the hybrid of DNA-RNA (*Figure 3B*). Polyacrylamide TBE-urea gel analyses of the resultant mixtures indicated that EXOSC1 notably cleaved ssDNA, while it displayed no detectable capability to cleave dsDNA or DNA-RNA hybrid (*Figure 3C*).

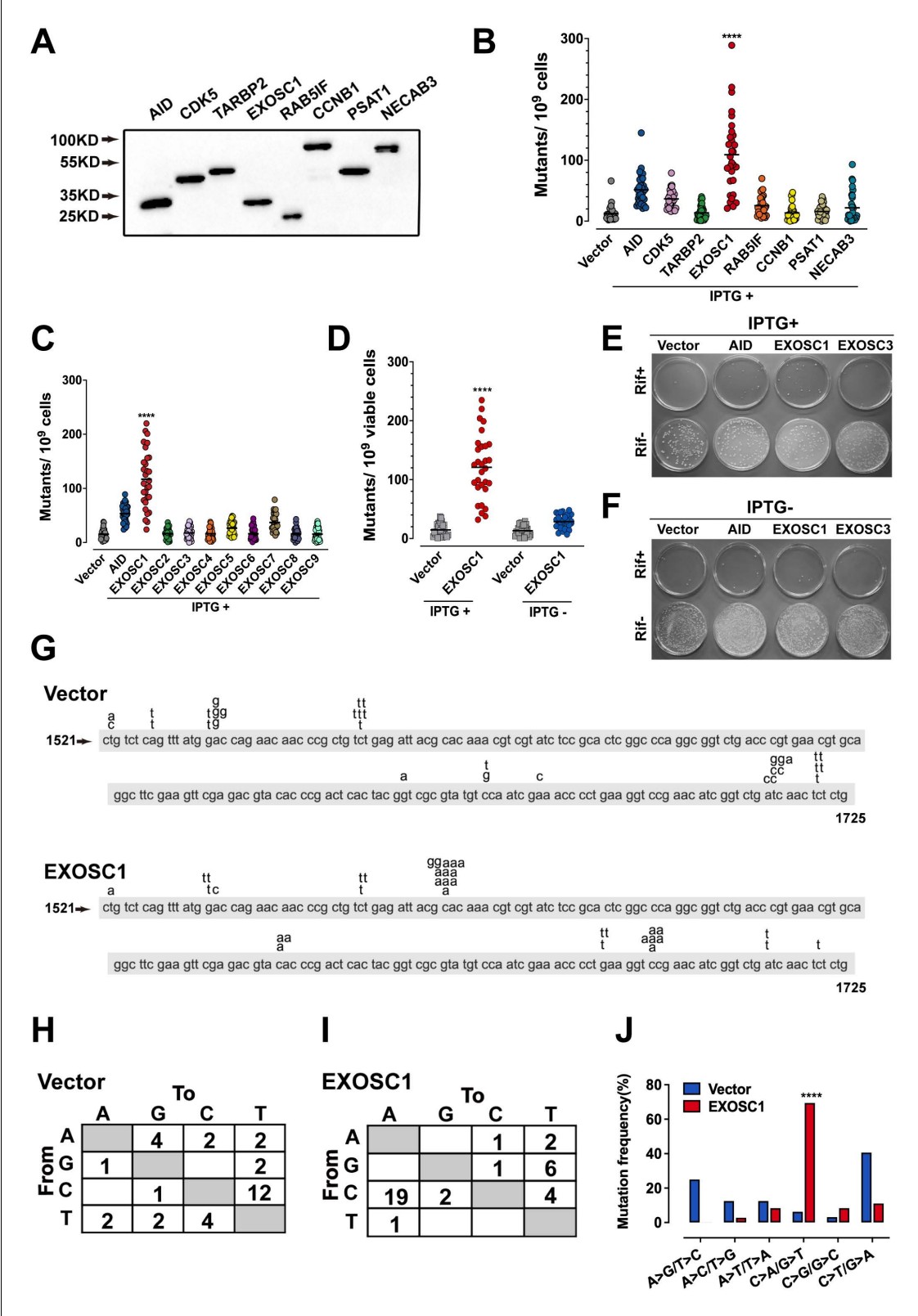

**Figure 2.** EXOSC1 promotes mutations in *Escherichia coli*. (**A**) Western blot showing His-tagged protein levels in *E. coli*. (**B, C**) The frequencies of the Rif[R] mutation in the *E. coli* cells expressing candidate ESMs (**B**) or exosome family members (**C**). The vector and AID were used as negative and positive controls, respectively. Each plot represents the mutational frequency of an independent overnight culture (n=30). Median mutational frequency of the gene is noted. (**D**) Frequencies of the Rif[R] mutation in the *E. coli* cells treated with and without IPTG (n=30). (**E, F**) Representative images of *E. coli* cells

*Figure 2 continued on next page*

*Figure 2 continued*

treated with (**E**) and without (**F**) IPTG. (**G**) The mutational distribution in 25 independent Rif$^R$ colonies transformed by vector or EXOSC1. (**H, I**) Summary of the c-substitutions in Rif$^R$ colonies transformed by vector (**H**) and EXOSC1 (**I**). (**J**) The mutational frequencies of each c-substitution in Rif$^R$ colonies. The p value was calculated by Fisher's exact test.

The online version of this article includes the following source data and figure supplement(s) for figure 2:

**Source data 1.** EXOSC1 promotes mutations in *E. coli*.
**Figure supplement 1.** EXOSC1 promotes mutations in *Escherichia coli*.
**Figure supplement 1—source data 1.** EXOSC1 promotes mutations in *E. coli*.

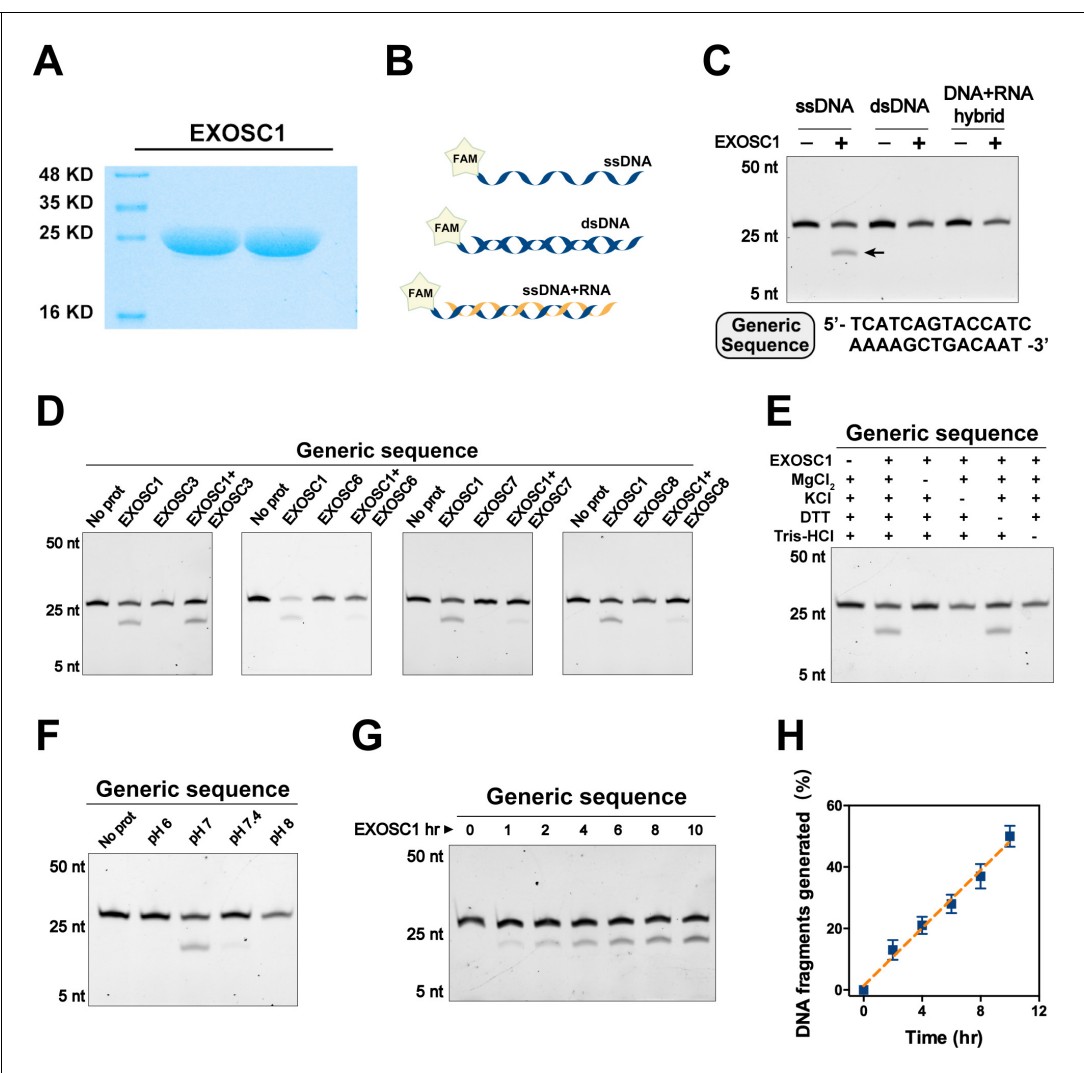

**Figure 3.** EXOSC1 cleaves ssDNA. (**A**) Coomassie blue staining of in vitro purified EXOSC1 protein. (**B**) Schematic of synthetic DNA substrates. (**C**) In vitro cleavage assays of EXOSC1 using generic ssDNA, dsDNA, and DNA-RNA hybrid as substrates. The assays were performed in the presence of 1 µM oligonucleotides, 1 µM EXOSC1, 70 mM KCl, 700 µM MgCl₂, 1 mM DTT, and 20 mM Tris–HCl pH 7.0. After incubation at 37°C for 4 hr, resultant samples were analyzed by 15% polyacrylamide TBE-urea gels. (**D**) Cleavage assays of EXOSC1 in the presence or absence of EXOSC3, EXOSC6, EXOSC7, and EXOSC8 using generic ssDNA as substrates. (**E**) Cleavage assays in the presence of the components as noted. (**F**) Cleavage assays of EXOSC1 at the pH as noted. (**G**) Time course cleavage assays of EXOSC1 using generic ssDNA as substrates. (**H**) Rate curve of EXOSC1 cleavage at 37°C and pH 7.0. dsDNA, double-stranded DNA; ssDNA, single-stranded DNA.

The online version of this article includes the following figure supplement(s) for figure 3:

**Figure supplement 1.** EXOSC1 cleaves ssDNA.

We then evaluated the capabilities of EXOSC1 homologs (EXOSC2–EXOSC9) to cleave ssDNA. The EXOSC protein was separately incubated with ssDNA. Gel analyses of the resultant mixtures indicated that, unlike EXOSC1, none of EXOSC2–EXOSC9 detectably cleaved ssDNA (*Figure 3—figure supplement 1A*). Considering that EXOSC1 is well known to form a complex with other exosome members, we also assessed the impact of the exosome members on the cleavage activity of EXOSC1. EXOSC1 was incubated with ssDNA in the presence of individual EXOSC1 homolog. Interestingly, EXOSC6, EXOSC7, and EXOSC8 decreased EXOSC1 cleavage activity (*Figure 3D*), while EXOSC2, EXOSC3, EXOSC4, and EXOSC9 showed no detectable impact (*Figure 3—figure supplement 1B*). And EXOSC1–9 (the exosome complex) displayed not detect DNA cleavage activity (*Figure 3—figure supplement 1C*). We then evaluated the impact of the reaction components and pH on the cleavage activity of EXOSC1. It was found that $K^+$ and $Mg^{2+}$ enhanced the cleavage activity of EXOSC1 (*Figure 3E*), and EXOSC1 showed the highest cleavage activity at pH 7.0 (*Figure 3F*). Further analyses indicated that the cleavage rate of EXOSC1 was approximately $4 \times 10^{-4}$/min at 37°C (*Figure 3G and H*).

## EXOSC1 prefers to cleave C sites in single-stranded DNA

Considering that the conserved exosome prefers to degrade the RNA with specific sequence (*Cvetkovic et al., 2017*), we determined whether EXOSC1 preferred to cleave some site(s) in ssDNA. EXOSC1 was incubated with DNAs containing unbiased sequence and distinct 3′ end. Consistent with the result of the generic DNA, EXOSC1 cleaved unbiased ssDNA and displayed no detectable capability to cleave dsDNA or DNA-RNA hybrid (*Figure 4A* and *Figure 4—figure supplement 1A*). And only EXOSC1 displayed cleavage activity against unbiased ssDNA (*Figure 4B*). Interestingly, the cleavage rate against unbiased ssDNA (approximately $1.2 \times 10^{-3}$/min) was higher than that against generic ssDNA (*Figure 4—figure supplement 1B*), suggesting that the DNA sequence might show some impact on the activity of EXOSC1. Notably, mass spectrometry (MS) analyses (*Figure 4—figure supplement 1C and D*) demonstrated that the resultant mixtures contained more free C than the other three base types (*Figure 4E*), suggesting that EXOSC1 preferred to cleave C sites in ssDNA. Consistently, EXOSC1 cleaved the ssDNA containing C>A hot spot of VHL mutation (*Figure 4—figure supplement 1E*). Since EXOSC1 was correlated with the C>A/G>T c-substitution type, it was likely that EXOSC1 cleaved C sites in ssDNA and subsequently resulted in C>A mutations through 'A' rule DNA repair.

To evaluate the above hypothesis, we then determined whether EXOSC1-promoted mutations displayed strand asymmetries. Considering that EXOSC1 cleaved C sites in ssDNA but not DNA-RNA hybrid, we speculated that 'transcribed' temple strands likely bound by RNA were less cleaved by EXOSC1. As shown in *Figure 4F*, C>A transversions in the 'untranscribed' coding strand lead to C>A mutations in a gene, while C>A transversions in the 'transcribed' template strand result in G>T mutations. Therefore, the capability of EXOSC1 to promote strand mutational asymmetry can be evaluated by comparing C>A and G>T frequencies. We first analyzed the distributions of C>A and G>T substitutions, instead of the distributions of C>A/G>T c-substitution, in the EXOSC1-transformed *E. coli* cells described above. As shown in *Figure 4—figure supplement 1F and G*, C>A substitution in rpoB gene was generated at a much higher frequency than G>T. Compared with the control, EXOSC1 enhanced C>A from 0% to 69% (p=$2.67 \times 10^{-7}$), while EXOSC1 only enhanced G>T from 6% to 17% even without a significance (p=0.27) (*Figure 4—figure supplement 1F and G*, and *Figure 4—figure supplement 1—source data 1*). Next, we evaluated C>A strand asymmetry in KIRC using Spearman's rank and Student's *t*-test analyses. Spearman's rank analyses indicated that EXOSC1 showed the highest correlation (r) with C>A, and the correlation between EXOSC1 and G>T was even lower than that between EXOSC1 and C>A/G>T (*Figure 4G*, *Figure 4—source data 1*, *Figure 4—figure supplement 1H*, and *Figure 4—figure supplement 1—source data 1*). To evaluate the impact of group number on the further Student's *t*-test analyses, the KIRC patients were grouped into two, three, four, or five groups according to the mutation types C>A, G>T, C>A/G>T and total (12 substitution types). As expected, the EXOSC1 differences between the low and high C>A/G>T groups were more significant than those between the low and high groups of total (12 substitution types) mutations (*Figure 4—figure supplement 1I and J*). Importantly, EXOSC1 showed more significant expression differences between the low and high C>A groups than those between the low and high G>T groups (*Figure 4H* and *Figure 4—figure supplement 1K*), suggesting that EXOSC1 prefers to cleave C sites in ssDNA in vivo.

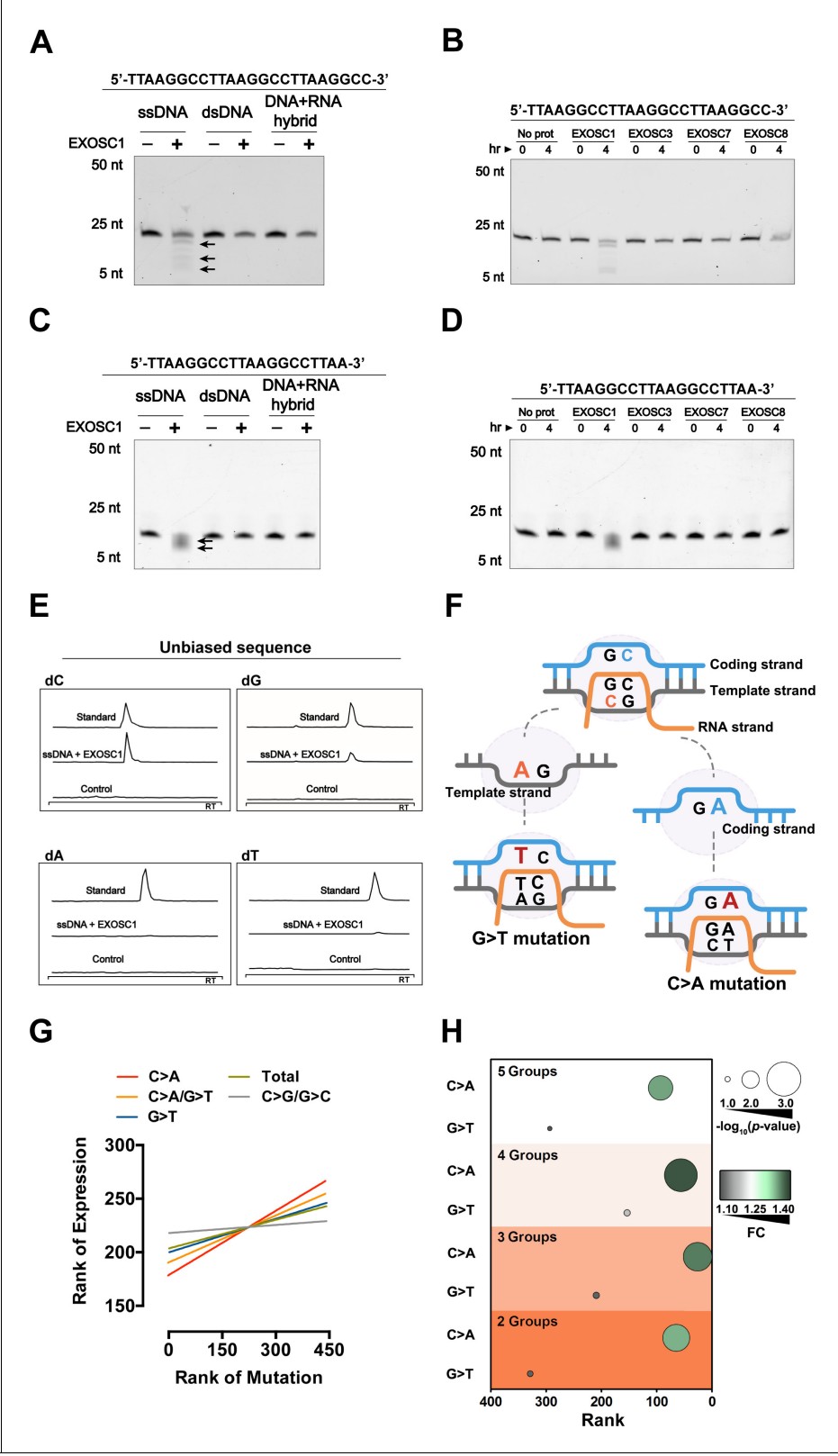

**Figure 4.** EXOSC1 prefers to cleave C sites in ssDNA. (**A, B**) Cleavage assay of EXOSC1 using unbiased DNA, dsDNA, and DNA-RNA hybrid as substrates. (**C, D**) Cleavage assays of EXOSC1, EXOSC3, EXOSC7 and EXOSC8 using unbiased ssDNA as substrates. (**E**) Mass spectrometry (MS) analyses of the resultant mixtures described in (**B**). C, G, A, and T were detected by MS using nucleoside to base ion mass transitions of 228.1–112.2 (**C**), 268.1–152.1 (**G**), 252.2–136.1 (**A**), and 243.1–127.2 (**T**), respectively (**Figure 3—figure supplement 1E**). Standard curves were generated by a serial

*Figure 4 continued on next page*

*Figure 4 continued*

dilution of C, G, A, and T (*Figure 3—figure supplement 1F*). Free C, G, A, and T contained in the reaction mixtures were quantified by standard curves. (F) Schematic showing that the consequence of C>A mutation in the coding strand is distinct from that in the template strand. (G) Correlation between EXOSC1 expression and the c-substitution mutation as noted in KIRC. Each line represents one best fit for visualization. P values were from Spearman's rank correlation. P and r values of C>A mutations were 0.0001 and 0.19, respectively; C>A/G>T: p=0.0006, r=0.17; G>T: p=0.0271, r=0.10; total mutations: p=0.0594, r=0.09; C/G>G/C: p=0.5730, r=0.03. (H) Student's *t*-test analyses of the expression difference of EXOSC1 between the high and low mutation groups. 2, 3, 4, and 5 represent the group numbers. For example, 4: the KIRC patients were grouped into four groups and the expression difference of EXOSC1 between the high and low mutation groups was analyzed by Student's *t*-tests. FC = mean gene expression in the high group/that in the low group. dsDNA, double-stranded DNA; KIRC, kidney renal clear cell carcinoma; ssDNA, single-stranded DNA.

The online version of this article includes the following source data and figure supplement(s) for figure 4:

**Source data 1.** EXOSC1 prefers to cleave C sites in ssDNA.

**Figure supplement 1.** EXOSC1 prefers to cleave C sites in ssDNA.

**Figure supplement 1—source data 1.** EXOSC1 prefers to cleave C sites in ssDNA.

## EXOSC1 enhances DNA damage and mutations in KIRC cells

Considering that EXOSC1 cleaves DNA in vitro, we then evaluated the capability of EXOSC1 to promote intracellular DNA damage using γ-H2AX staining and neutral comet tail assays. The 769 P and TUHR14TKB KIRC cells were transfected with the plasmid encoding EXOSC1 using empty vector as a control (*Figure 5—figure supplement 1A*). γ-H2AX staining analyses of the resultant cells demonstrated that EXOSC1 increased the γ-H2AX foci in the cells (*Figure 5A*). The number of γ-H2AX-positive cells was increased approximately sevenfold by EXOSC1 (*Figure 5B* and *Figure 5—source data 1*). While knockdown of EXOSC1 reduced the γ-H2AX foci (*Figure 5C*, *Figure 5—source data 1*, *Figure 5—figure supplement 1B and C*, and *Figure 5—figure supplement 1—source data 1*). Consistent with the results of γ-H2AX staining, comet tail analyses also indicated that EXOSC1 increased DNA damage (*Figure 5D and E*, and *Figure 5—source data 1*).

Due to the central role of DNA damage in mutations, we performed differential DNA denaturation PCR (3D-PCR) to determine whether EXOSC1 enhances mutation in KIRC cells. Because that DNA sequences with more A/T content can be amplified at lower denaturation temperatures than parental sequences, 3D-PCR enables qualitative estimates of genomic C/G>A/T mutations in a population of cells. As shown in *Figure 5F*, the enhanced expression of EXOSC1 (EXOSC1-OE) increased the lower temperature amplicons (LTAs) of VHL, suggesting that EXOSC1 increased the mutations in VHL gene. Consistently, further sequencing analyses of the LTAs indicated that EXOSC1-OE cells showed more C>A mutations in VHL gene (*Figure 5F*).

Considering that the 'A' rule DNA repair is dependent on X-ray repair cross-complementing 1 (XRCC1) (*Sale et al., 2001*), we knocked down XRCC1 to evaluate the role of XRCC1 in EXOSC1-promoted mutations (*Figure 5—figure supplement 1D-F*, and *Figure 5—figure supplement 1—source data 1*). 3D-PCR analyses of the resultant cells indicated that knockdown of XRCC1 impaired the capability of EXOSC1 to increase the LTAs (*Figure 5G*). Furthermore, both XRCC1 knockdown (XRCC1-KD) and EXOSC1 knockdown (EXOSC1-KD) decreased the LTAs (*Figure 5H and I*). Additionally, a subcutaneous xenograft tumor model was used to determine whether EXOSC1 enhances DNA mutations in vivo (*Figure 5J*). Stable control (vector), EXOSC1-OE, and EXOSC1-KD 769 P cells were subcutaneously implanted. After 2 weeks, 3D-PCR analyses of the resultant tumors indicated that EXOSC1 increased the LTAs of VHL, whereas knockdown of EXOSC1 reduced the LTAs (*Figure 5K*), suggesting that EXOSC1 enhanced mutations in KIRC.

## EXOSC1 sensitizes KIRC cells to PARP inhibitor

Considering the central roles of mutation in the process of cancers, we evaluated the potential clinical significance of EXOSC1 in KIRC using KM analyses. KM analyses of disease-free survival (DFS) and OS were performed using the clinical data from 532 KIRC patients in TCGA. The fragments per kilobase per million mapped reads (FPKMs) were used to evaluate the expression of EXOSC1 in KIRC (*Figure 6—figure supplement 1A*, and *Figure 6—figure supplement 1—source data 1*). The median-separation KM analyses indicated that high EXOSC1 group showed poor DFS and OS (*Figure 6A and B*, and *Figure 6—source data 1*). The median DFS in the low EXOSC1 group was 32.0 months longer than that in the high group (p=9.78 $\times$ 10$^{-8}$, log-rank test) (*Figure 6A* and

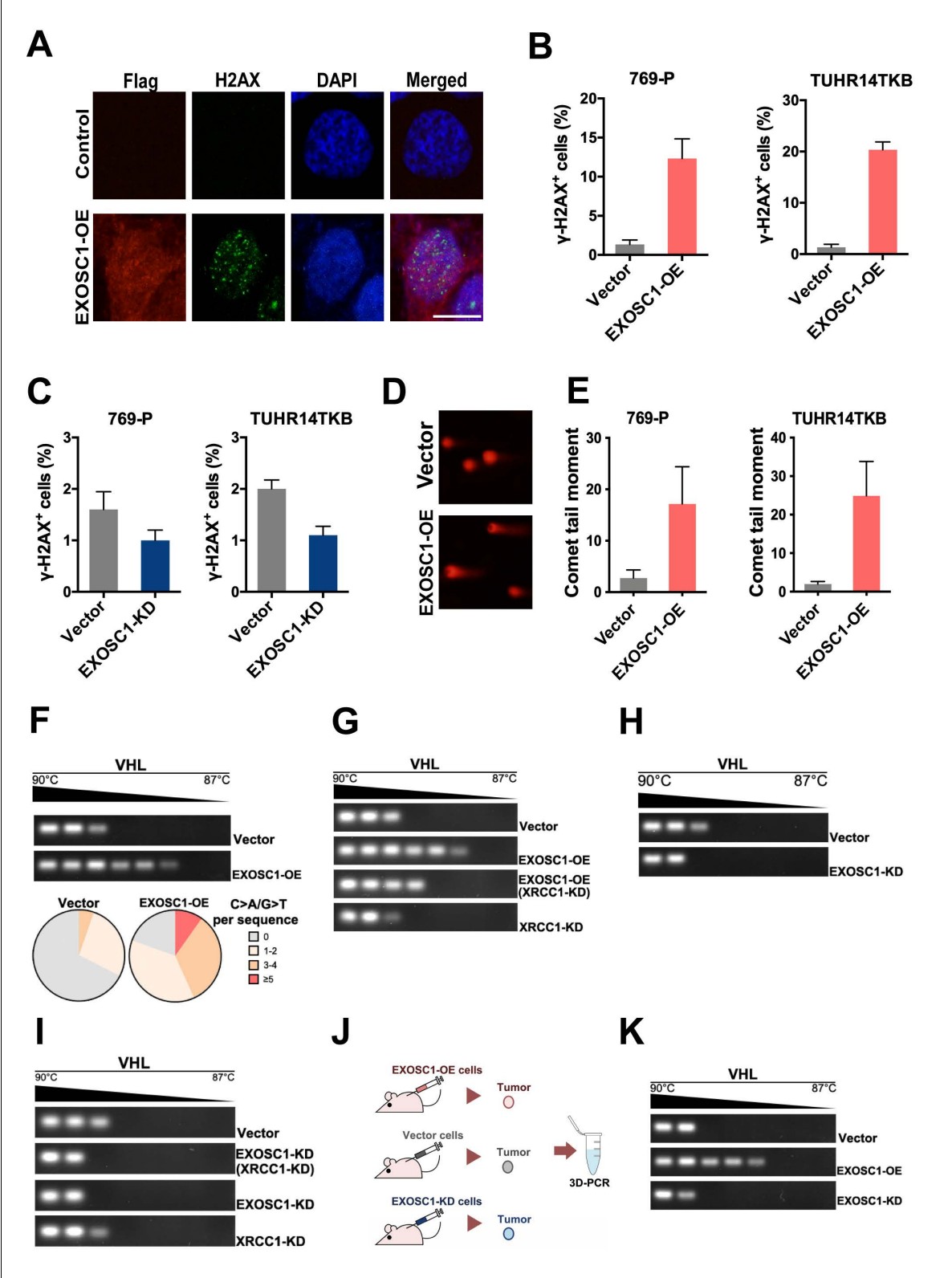

**Figure 5.** EXOSC1 enhances DNA damages and mutations in KIRC Cells. (**A**) Representative fluorescent images of γ-H2AX foci in 769 P cells transfected with control (pCDH, empty vector) or pCDH-Flag EXOSC1 plasmids for 2 days. Scale bar=10 μm. (**B**) Percentage of cells with more than 20 γ-H2AX foci in the KIRC cells transfected with control or pCDH-Flag EXOSC1 plasmids for 2 days. (**C**) Percentage of cells with more than 20 γ-H2AX foci in the KIRC cells infected with lentivirus encoding shRNAi control (pLKO scramble) or pLKO sh-EXOSC1 for 2 days. (**D**) Representative images of DNA

*Figure 5 continued on next page*

*Figure 5 continued*

comets in 769 P cells transfected with control or pCDH-Flag EXOSC1 plasmids for 2 days. (E) Comet tail moment of the 769 P and TUHR14TKB cells transfected with control or pCDH-Flag EXOSC1 plasmids for 2 days. (F) 3D-PCR and subsequent sequencing analyses of the VHL mutations in the TUHR14TKB cells stably expressing control (vector) or EXOSC1 (pCDH-Flag EXOSC1, EXOSC1-OE). (G) 3D-PCR analyses of VHL in TUHR14TKB cells stably expressing control (pLKO.1 vector) or shRNA against EXOSC1 (pLKO shEXOSC1-1, EXOSC1-KD). (H) 3D-PCR analyses of VHL in TUHR14TKB cells stably expressing control, EXOSC1-OE and/or shRNA against XRCC1 (pMKO.1 shXRCC1-1, XRCC1-KD). (I) 3D-PCR analyses of VHL in TUHR14TKB cells stably expressing control (pLKO.1 vector), shRNA against EXOSC1 (pLKO shEXOSC1-1, EXOSC1-KD) and/or shRNA against XRCC1 (pMKO.1 shXRCC1-1, XRCC1-KD). (J) Schematic showing the subcutaneous xenograft tumor models. The $5 \times 10^6$ control, EXOSC1-OE and EXOSC1-KD 769 P cells were implanted subcutaneously. After 2 weeks, the resultant tumors were analyzed by 3D-PCR. (K) 3D-PCR analyses of VHL in the tumors described in (J). KIRC, kidney renal clear cell carcinoma.

The online version of this article includes the following source data and figure supplement(s) for figure 5:

**Source data 1.** EXOSC1 enhances DNA damages and mutations in KIRC cells.

**Figure supplement 1.** EXOSC1 enhances DNA damages and mutations in KIRC cells.

**Figure supplement 1—source data 1.** EXOSC1 enhances DNA damages and mutations in KIRC cells.

*Figure 6—source data 1*). Consistently, the median OS in the low EXOSC1 group was 36.9 months longer than that in the high group (p=2.2 × 10$^{-8}$) (*Figure 6B* and *Figure 6—source data 1*). As expected, the best-separation KM analysis also indicated that high EXOSC1 group significantly showed poor OS (p=2.6 × 10$^{-12}$) (*Figure 6—figure supplement 1B*, and *Figure 6—figure supplement 1—source data 1*). Due to the critical role of VHL mutation in KIRC, we then evaluated the potential clinical significance of EXOSC1 in the presence and absence of VHL mutation. KM analyses indicated that high EXOSC1 group showed poor OS in both presence (median OS [high vs. low group]=65.2 vs. 98.5 months, p=0.015) and absence (median OS [high vs. low group]=75.7 vs. 104.5 months, p=1.0 × 10$^{-5}$) of VHL mutation (*Figure 6C*, *Figure 6—source data 1*, and *Figure 6—figure supplement 1C*). Furthermore, median separation KM analyses indicated that OS was not significantly different based on EXOC1 expression in BRCA, GBM, BLCA, and AML patients (*Figure 6—figure supplement 1D* and *Figure 6—figure supplement 1—source data 1*), suggesting a role of EXOC1 in the treatment against KIRC.

Considering that EXOSC1 increases DNA damage, we speculated that EXOSC1 potentially sensitizes KIRC cells to the inhibitors of poly(ADP-ribose) polymerase (PARP), which treats cancers via blocking DNA repair. As previously described (*Li et al., 2020*), colony formation assays were performed to evaluate the role of EXOSC1 in response to the PARP inhibitors, niraparib, and olaparib. Stable control (vector) and enhanced EXOSC1 (EXOSC1-OE) KIRC cells were seeded and treated with serial dilutions of PARP inhibitor until colonies were notably formed. As expected, both niraparib and olaparib more notably inhibited the KIRC cells with enhanced EXOSC1 (*Figure 6D–F*, *Figure 6—source data 1*, *Figure 6—figure supplement 1E and F*, and *Figure 6—figure supplement 1—source data 1*), suggesting that EXOSC1 sensitized the cells to the PARP inhibitor. Next, we determined whether EXOSC1 could sensitize KIRC cells to niraparib in xenograft mouse models. The control and EXOSC1-OE cells were subcutaneously injected. Resultant tumor-bearing mice were grouped and treated by vehicle or niraparib. Consistent with the ex vivo results, niraparib more notably inhibited the tumor with enhanced EXOSC1 (*Figure 6G and I*, and *Figure 6—source data 1*), indicating that EXOSC1 sensitized KIRC xenografts to the inhibitor. No significant weight loss was observed throughout the study, suggesting that the niraparib treatment was well tolerated (*Figure 6H and J*, and *Figure 6—source data 1*).

## Discussion

The genomic integrity of human cells is constantly assaulted by ESMs. Although human cells possess multi DNA repair mechanisms to counteract these constant assaults, not all lesions are correctly repaired and almost inevitably result in mutations. The central roles of these acquired mutations in nearly all cancers (*Jeggo et al., 2016*; *Roos et al., 2016*) emphasize the identification and understanding of the ESMs. Here, we show that EXOSC1 cleaves ssDNA and acts as an ESM in KIRC. Consistent with the capability of EXOSC1 to promote DNA damage and mutations, KIRC patients with high EXOSC1 showed a poor prognosis, and EXOSC1 also sensitized cancer cells to the PARP inhibitor.

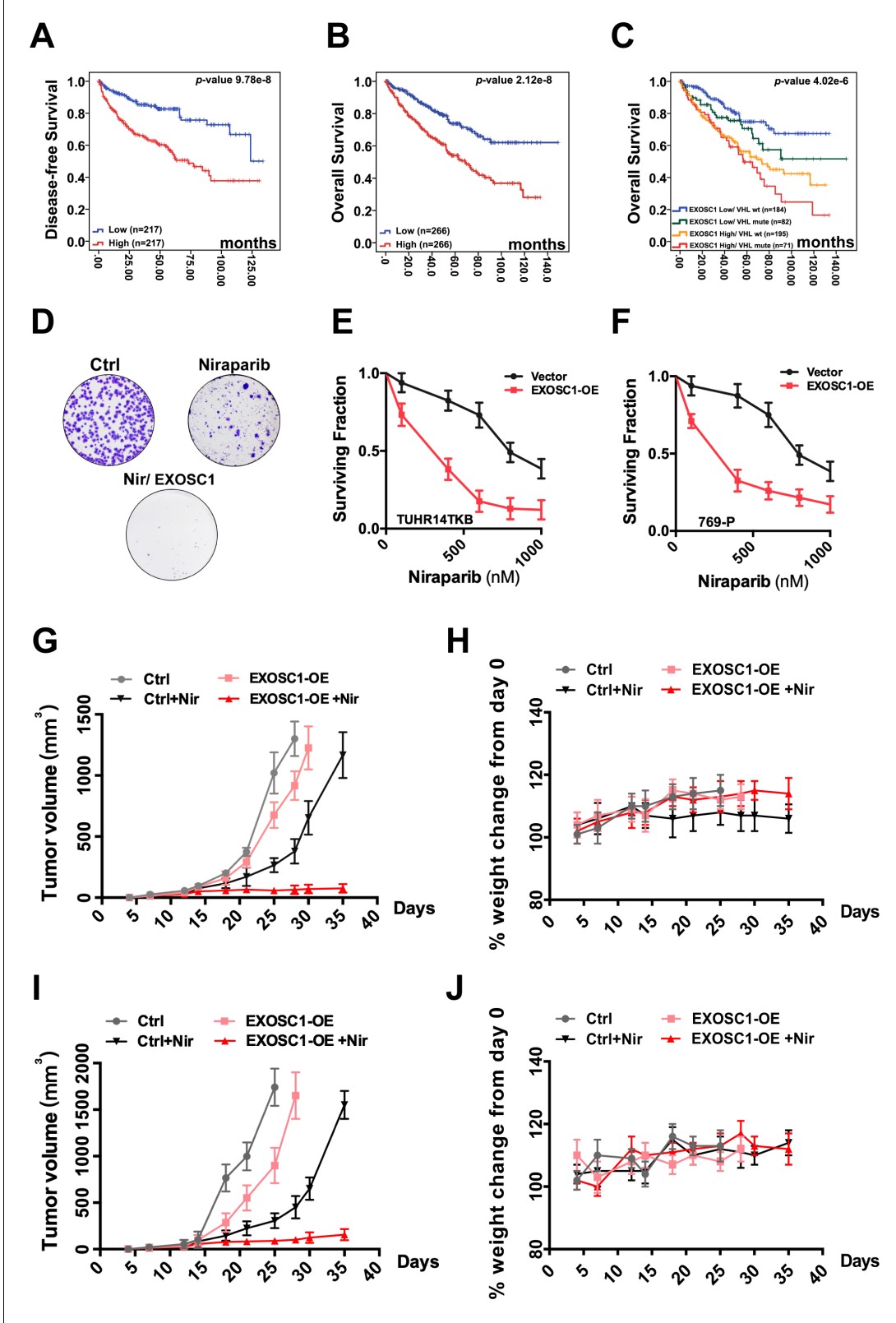

**Figure 6.** EXOSC1 sensitizes KIRC cells to PARP inhibitor. (**A, B**) KM analyses of DFS (**A**) and OS (**B**) in KIRC patients with different EXOSC1 levels. P values were obtained from the log-rank test. (**C**) KM analysis of OS in KIRC patients with different EXOSC1 expression levels and VHL mutations. (**D**) Colony formation of control and EXOSC1-OE TUHR14TKB cells treated with 600 nM niraparib. (**E, F**) Clonogenic survival of control and EXOSC1-OE TUHR14TKB (**E**) and 769 P (**F**) cells treated with niraparib. (**G, I**) Tumor volumes of 769 P (**G**) and Caki-2 (**I**) xenografts treated with or without niraparib

*Figure 6 continued on next page*

*Figure 6 continued*

(n=4 groups; n=6 mice/group; ± SEM). (**H, J**) Body weight change percentage of 769 P (**H**) and Caki-2 (**J**) xenografts treated as described in (**G**). DFS, disease-free survival; KIRC, kidney renal clear cell carcinoma; KM, Kaplan-Meier; OS, overall survival.

The online version of this article includes the following source data and figure supplement(s) for figure 6:

**Source data 1.** EXOSC1 sensitizes KIRC cells to PARP inhibitor.
**Figure supplement 1.** EXOSC1 sensitizes KIRC cells to PARP inhibitor.
**Figure supplement 1—source data 1.** EXOSC1 sensitizes KIRC cells to PARP inhibitor.

Our results show that a unit of multiprotein complex can play a role distinct from the function(s) of the complex. Biological processes frequently require fine control over the formation of a multiprotein complex in a particular region of the cell. The exosome complex is well known for its roles in RNA degradation (*Januszyk and Lima, 2014*; *Kilchert et al., 2016*). However, the role of exosome complex members other than RNA binding and degradation remains elusive. Interestingly, EXOSC1 can disassociate from the exosome complex (*Dai et al., 2018*; *Malet et al., 2010*), suggesting that EXOSC1 might be involved in some functions independent of the exosome. Our study demonstrated that EXOSC1 acts as an ESM to promote mutagenesis. Conversely, previous studies have described that the exosome, as a multiprotein complex, protects cells from genomic instability by preventing the formation of DNA/RNA hybrids and restricting DNA strand mutational asymmetry (*Lim et al., 2017*; *Pefanis and Basu, 2015*; *Pefanis et al., 2015*). This phenomenon can at least partially be explained by the finding that some exosome members (EXOSC7 and EXOSC8) can block the activity of EXOSC1 to cleave DNA. Therefore, a single EXOSC1 protein can show different responses depending on the presence/absence of its interacting partners. Further studies are needed to better understand the roles of the individual exosome member.

The potential pathological significance of EXOSC1 is supported by its association with poor DFS and OS in KIRC. Due to the capability of EXOSC1 to cleave DNA and promote mutations, EXOSC1 might enhance mutations and consequently provide genetic fuel for cancer development, metastasis, and even therapy resistance. Therefore, EXOSC1 might represent not only a KIRC marker but also a target to decrease the rate of KIRC evolution and stabilize the targets of existing therapeutics. Furthermore, targeting DNA repair in cancers by inhibiting PARPs offers an important therapeutic strategy (*Cleary et al., 2020*). Unfortunately, the failure of PARP inhibitors to markedly benefit patients enforces the necessity for developing new strategies to improve their efficacy (*Cleary et al., 2020*; *Dizon, 2017*; *Lord and Ashworth, 2017*). Our study demonstrated that EXOSC1 sensitized KIRC cells to PARP inhibitor, suggesting inhibition of PARPs might be a penitential strategy to treat KIRC patients with high EXOSC1. We also noticed that KIRC patients with high EXOSC1 and VHL mutations showed the poorest OS. Considering the DNA cleavage activity of EXOSC1 and the role of VHL in stabilizing the genome (*Thoma et al., 2009*; *Zhang et al., 2018*), we speculate that patients with the VHL mutation and high EXOSC1might show higher possibility to benefit from PARP inhibitor(s).

However, several limitations of this study should be noted. First, we observed a notable variation in terms of the correlation with a different c-substitution type for a given gene, implying the need for further studies. Second, although we showed that EXOSC1 could cleave ssDNA and act as an ESM, we did not directly identify the mechanism responsible for turning the DNA cleavages into mutations. The role of XRCC1 in EXOSC1-promoted mutations was only briefly evaluated. Hence, we cannot exclude the possibility that other proteins might contribute to this process. Third, the candidate ESM genes in KIRC showed notable enrichment in 'mitochondrial gene expression' and 'organophosphate biosynthetic process', suggesting a role of lipid metabolism. Unfortunately, due to the limitation of in vitro rifampicin-resistant assay, we did not focus on the lipid metabolism in this study. Since that KIRC cells are well known to contain many lipid droplets, it will be interesting to determine whether and how lipid metabolism acts as an ESM. Despite these limitations, our results still indicate that EXOSC1 acts as an ESM in KIRC.

## Materials and methods

### Key resources table

| Reagent type (species) or resource | Designation | Source or reference | Identifiers | Additional information |
|---|---|---|---|---|
| Antibody | Anti-Flag (rabbit polyclonal) | Sigma-Aldrich | Cat. number F7425 (RRID:AB_439687) | Western blot (1:1000) |
| Antibody | Anti-His (rabbit polyclonal) | Sigma-Aldrich | Cat. number SAB1306085 | Western blot (1:1000) |
| Antibody | Anti-phospho-γ-H2AX (Ser139) (mouse monoclonal) | Millipore | Cat. number 05–636 (RRID:AB_309864) | IF (1:200) |
| Antibody | Anti-XRCC1 (rabbit polyclonal) | Millipore | Cat. number ABE559 | Western blot (1:1000) |
| Antibody | Anti-EXOSC1 (rabbit monoclonal) | Abcam | Cat. number EPR13526 | Western blot (1:1000) |
| Chemical compound, drug | Niraparib | MedChem Express | Cat. number HY-10619 | |
| Chemical compound, drug | Olaparib | MedChem Express | Cat. number HY-10162 | |
| Chemical compound, drug | Rifampicin | Sigma-Aldrich | Cat. number R3501 | |
| Software, algorithm | Prism | GraphPad | Version 8 | data analyses |
| Other | Lipofectamine 2000 | Thermo Fisher Scientific Inc | Cat. number 11668019 | |

## Sample preparation

Samples of 532 KIRC patients from TCGA used for expression and mutation analyses were collected through The cBio Cancer Genomics Portal (http://cbioportal.org) as described in our previous studies (Li et al., 2020; Zhou et al., 2019).

## Cell culture

All cell lines were obtained from the American Type Tissue Collection. The 769 P, SNU-1272, and Caki-2 cells were maintained in RPMI 1640 medium containing 10% heat-inactivated fetal bovine serum (FBS), 100 U/ml penicillin, and streptomycin at 37°C under a humidified atmosphere of 5% $CO_2$ Dulbecco's modified Eagle's medium containing 10% heat-inactivated FBS, 100 U/ml penicillin, and streptomycin at 37°C under a humidified atmosphere of 5% $CO_2$. Mycoplasma PCR testing of these cells was performed every week, and no mycoplasma was detected. Transfections were performed using lipofectamine 2000 (Thermo Fisher Scientific Inc, Waltham, MA).

## Antibodies, reagents, and plasmids

Anti-Flag (cat. number F7425) and anti-His (cat. number SAB1306085) antibodies were from Sigma-Aldrich (St. Louis, MS). Anti-phospho-γ-H2AX (Ser139) (cat. number 05-636) and anti-XRCC1 (cat. number ABE559) were from Millipore (Billerica, MA). Anti-EXOSC1 (cat. number EPR13526) was from Abcam (Cambridge, MA). Niraparib (cat. number HY-10619) and olaparib (cat. number HY-10162) were from MedChem Express. Rifampicin (cat. number R3501) was from Sigma-Aldrich.

Full-length EXOSC1 was cloned into the Xba I and Nhe I sites of the lentivirus vector pCDH-CMV-MCS-EF1-Puro (System Biosciences, Mountain View, CA) to construct pCDH-Flag EXOSC1. CCGAG TTCCTACAGACCTAAG and CGAGGAACTATCCGCAAAGAA sequences were cloned into pLKO.1 to construct the pLKO shEXOSC1-1 and pLKO shEXOSC1-2 plasmids, respectively. Similarly, CCAG TGCTCCAGGAAGATATA and CGATACGTCACAGCCTTCAAT sequences were cloned into pMKO.1 to construct the pMKO shXRCC1-1 and pMKO shXRCC1-2 plasmids, respectively. According to the knockdown efficiency (Figure 5—figure supplement 1), pLKO shEXOSC1-1 and pMKO.1 shXRCC1-1 with higher knockdown efficiencies were used to generate EXOSC1-KD and XRCC1-KD cells. AID (NM_020661.4), CDK5 (NM_004935.4), TARBP2 (NM_134323.1), EXOSC1 (NM_016046.5), RAB5IF (NM_018840.5), CCNB1 (NM_031966.4), PSAT1 (NM_058179.4), NECAB3 (NM_031232.3),

EXOSC2 (NM_014285.7), EXOSC3 (NM_016042.4), EXOSC4 (NM_019037.3), EXOSC5 (NM_020158.4), EXOSC6 (NM_058219.3), EXOSC7 (NM_015004.4), EXOSC8 (NM_181503.3), and EXOSC9 (NM_005033) were amplified and cloned into the pET-28a(+) vector (Novogen Limited, Hornsby Westfield, NSW) to construct the pET-28a-Gene-6XHis *E. coli* expression plasmids. PCR primers for the amplification of the above genes are described in *Supplementary file 5*.

## Immunoblotting and immunofluorescence

Immunoblotting and immunofluorescence were carried out as described in our previous study (*Song et al., 2018*; *Wang et al., 2020*).

## Rifampicin-resistant assay in *E. coli*

Rifampicin-resistant assays were carried out as described previously (*Petersen-Mahrt et al., 2002*). Briefly, rifampicin-resistant assays for each gene were performed using 30 independent cultures grown overnight to saturation in a rich medium supplemented with 50 mg/L kanamycin and 1 mM IPTG. $Rif^R$ mutants were selected on a medium containing 50 mg/L rifampicin. Mutation frequencies were assessed by determining the median number of rifampicin-resistant clones per $10^9$ viable plated cells. The mutation spectra of $Rif^R$ were analyzed by sequencing the amplified rpoB 627 bp PCR products using 5′-TTGGCGAAATGGCGGAAAACC-3′ and 5′-CACCGACGGATACCACCTGC TG-3′ primers.

## Expression and purification of EXOSC proteins

Expression and purification of EXOSC proteins were carried out as described in our previous study (*Wang et al., 2020*). pET-28a-EXOSCs-6XHises encoding His-tagged EXOSCs were introduced into BL21 (DE3)-pLysS, which were grown in nutrient-rich medium with 32Y (containing 3.2% (w/v) yeast extract, 0.8% (w/v) peptone, and 0.58% (w/v) NaCl) in 10 mM Tris–HCl at 30°C and pH 7.4. Protein expression was induced with 0.4 mM IPTG at 20°C for 20 hr after the cells reached an $OD_{600}$ of 0.4–0.5. Induced BL21 (DE3)-pLysS host cells without any plasmid were used as a negative control. The resultant cells were harvested by centrifugation at $5000 \times g$ for 10 min and washed ttwice with ice-cold phosphate-buffered saline (PBS). The collected cells were resuspended in PBS (1 g of wet weight cells per 10 mL of PBS) containing 1 mM $MgCl_2$, 20 mM imidazole, one tablet/50 mL protease inhibitor cocktail, and 100 U/mL DNase. Resuspended cells were broken by an ultrasonic wave. Cell lysates were centrifuged at $20,000 \times g$ at 4°C for 30 min to remove unbroken cells and debris.

After pre-equilibration with 10 column volumes (CVs) of binding buffer (PBS containing 10% (v/v) glycerol and 20 mM imidazole, pH 7.6), Ni sepharose 6 Fast flow (GE Healthcare, New York, NY) was applied for the purification of EXOSCs. The resins were washed five times and eluted using elution buffer (binding buffer containing 300 mM imidazole). EXOSCs were concentrated using an Amicon Ultrafree centrifugal filter (Millipore Corporation, Billerica, MA) and pre-equilibrated with 10 mM HEPES buffer (pH 7.4) containing 150 mM NaCl and 10% glycerol. Size-exclusion chromatography (SEC) with a Superdex-200 HiLoad 10/600 column was used to further purify EXOSCs. The purity of the fractions was analyzed by coomassie blue staining. The protein concentration was determined using a BCA assay according to the manufacturer's instructions (Pierce, Rockland, IL).

## Cleavage activity assay in vitro

The cleavage assays of EXOSCs were carried out in reaction buffer modified from a previous study (*Liu et al., 2006*). Briefly, 50 µL of reaction mixture containing 1 µM oligonucleotides, 1 µM EXOSC protein, 70 mM KCl, 700 µM $MgCl_2$, 1 mM DTT, and 20 mM Tris–HCl pH 7.0 was incubated at 37°C for 4 hr. The reaction was stopped by addition of 10 µM proteinase K at 58°C for 10 min and heating at 90°C for 30 s. The resultant samples were then analyzed using 15% polyacrylamide TBE-urea gels.

## LC-MS/MS analysis

LC-MS/MS analyses of deoxyadenine (A), deoxythymidine (T), deoxyguanine (G), and deoxycytocine (C) were carried out as described in our previous studies (*Song et al., 2018*; *Wang et al., 2020*).

## Generation of stable cell lines

Stable cell lines were generated as described in our previous study (*Song et al., 2018*; *Wang et al., 2020*). Briefly, the TUHR14TKB, SNU-1272, 769 P, and Caki-2 cells were infected with pCDH-CMV-MCS-EF1-Puro (empty vector used as control), pCDH-Flag EXOSC1 (EXOSC1-OE), pLKO.1-scramble shRNA (empty vector used as control), pLKO shEXOSC1-1 (EXOSC1-KD), or pLKO shEXOSC1-2 lentiviral particles, which were generated following the manufacturer's protocol (System Biosciences, Mountain View, CA). The resultant cells were selected with puromycin for 2 weeks. These stable cells were then infected with virus encoding pMKO.1, pMKO.1 shXRCC1-1 (XRCC1-KD), or pMKO.1 shXRCC1-2. The resultant cells were selected with hygromycin B for 2 weeks to generate stable XRCC1 knockdown cells. According to the knockdown efficiency (*Figure 5—figure supplement 1*), shEXOSC1-1 and shXRCC1-1 with higher knockdown efficiencies were used as EXOSC1-KD and XRCC1-KD in this study.

## γ-H2AX staining and neutral comet tail assays

γ-H2AX staining and neutral comet tail assays were performed as described previously (*Li et al., 2020*).

## 3D-PCR and sequencing

3D-PCRs of VHL mutations were carried out as described previously (*Burns et al., 2013a*) using first (5′-GAGTACGGCCCTGAAGAAGA-3′ and 5′-TCAATCTCCCATCCGTTGAT-3′) and nested (5′-TGCGCTAGGTGAACTCGC-3′ and 5′-GCGGCAGCGTTGGGTAGG-3′) PCR primers. PCR products were then analyzed by gel electrophoresis, cloned into pMD20-T vector, and sequenced.

## Colony-forming assay

The colony-forming assays were performed as described in our previous study (*Li et al., 2020*).

## Subcutaneous xenograft tumor growth in vivo

The following animal-handling procedures were approved by the Animal Care and Use Committee of Dalian Medical University. Xenograft models were carried out as described in our previous studies (*Li et al., 2020*; *Song et al., 2018*; *Yang et al., 2010*). Briefly, $2 \times 10^6$ stable control/EXOSC1-OE 769 P and Caki-2 cells were suspended and injected subcutaneously into the flank of 6-week-old nude mice. After 7 days, these tumor-bearing mice were randomized into four groups (six mice per group) and treated by oral gavage twice a day with vehicle or niraparib (4 mg/kg). The mice were observed daily and weighed once per week. Tumor size was measured using a caliper, and the tumor volume was calculated using the following formula: $0.52 \times L \times W^2$, where L is the longest diameter and W is the shortest diameter. Mice were euthanized when the tumors reached 1500 mm$^3$ or showed necrosis.

## Statistical analyses

P values were calculated by the two-tailed Student's *t*-test, log-rank test, Fisher's exact test, chi-squared test, or Spearman's correlation analyses as noted. P values<0.05 were considered statistically significant.

## Experimental replicates and reproducibility

All data presented in this paper are representative of two to four independent experiments with comparable results.

## Acknowledgements

The authors would like to thank Dr. Wei Cheng (ICSC Core Facility, Dalian Medical University) for her material and technical support. Funding: This study was supported by grants from the National Natural Science Foundation of China (NSFC Nos. 81872310 to QY, and 82073123 to CS) and the China Postdoctoral Science Foundation (2020M680956 to LW). The project is sponsored by Liaoning Bai-QingWan Talents Program.

## Additional information

### Funding

| Funder | Grant reference number | Author |
|---|---|---|
| National Natural Science Foundation of China | 81872310 | Qingkai Yang |
| National Natural Science Foundation of China | 82073123 | Chengli Song |
| China Postdoctoral Science Foundation | 2020M680956 | Lina Wang |

The funders had no role in study design, data collection and interpretation, or the decision to submit the work for publication.

### Author contributions
Qiaoling Liu, Zhen Sun, Kai Wang, Data curation, Investigation; Qi Xiao, Na Wang, Investigation; Bo Wang, Data curation, Formal analysis, Investigation, Writing - review and editing; Lina Wang, Supervision, Validation, Investigation; Chengli Song, Data curation, Formal analysis, Funding acquisition, Investigation, Writing - original draft; Qingkai Yang, Supervision, Funding acquisition, Investigation, Methodology, Writing - original draft, Project administration, Writing - review and editing

### Author ORCIDs
Qingkai Yang [ID] https://orcid.org/0000-0001-6628-5393

### Decision letter and Author response
Decision letter https://doi.org/10.7554/eLife.69454.sa1
Author response https://doi.org/10.7554/eLife.69454.sa2

## Additional files

### Supplementary files
• Supplementary file 1. Table for the candidates identified by Spearman's rank analyses at whole genomic level.

• Supplementary file 2. Table for the candidates identified by Student's *t*-test analyses at whole genomic level.

• Supplementary file 3. Table for the genes showing significant association with VHL.

• Supplementary file 4. Table for the overlap numbers of the candidate genes of the six types of c-substitutions.

• Supplementary file 5. Table for primer sequence.

• Transparent reporting form

### Data availability
All data associated with this study are available in the main text or the supplementary materials.

The following previously published dataset was used:

| Author(s) | Year | Dataset title | Dataset URL | Database and Identifier |
|---|---|---|---|---|
| Yang Q | 2013 | Comprehensive molecular characterization of clear cell renal cell carcinoma | http://www.cbioportal.org/study/summary?id=kirc_tcga_pub | cBioPortal, KIRC |

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
