## [Decision Letter]

**Acceptance summary:**

In this study, a novel molecule, EXOSC1 was proposed to be associated with DNA damage, and thus could serve as potential biomarker predicting efficacy of therapy in kidney cancer. The studies were rigorously designed and executed to illustrate the main points.

**Decision letter after peer review:**

Thank you for submitting your article "Exosome component 1 cleaves single-stranded DNA and sensitizes kidney renal clear cell carcinoma cells to poly(ADP-ribose) polymerase inhibitor" for consideration by *eLife*. Your article has been reviewed by 3 peer reviewers, one of whom is a member of our Board of Reviewing Editors, and the evaluation has been overseen by Mone Zaidi as the Senior Editor. The following individual involved in review of your submission has agreed to reveal their identity: Longtao Wu (Reviewer #3).

Essential revisions:

1) The amount of work was sufficient to justify the conclusion. However, validation through basic experiment and clinical data analysis still requires further improvement.

2) Data presentation in the figures could be adjusted more reasonable, and several questions regarding the figures remain to be explained for enriching the content and improving this work better.

3) Some points in the manuscript were not in accordance with the figures, please check and make corresponding adjustments.

*Reviewer #1 (Recommendations for the authors):*

1. In line 147-148: 'KIRC displayed higher frequencies of C>A/G>T, A>T/T>A, and A>C/T>G mutations', as is manifested in Figure 1I, most frequently mutated substitutions are C>T/G>A, which is not in accordance with the description.

2. In figure 6, the photographs on tumor mass from PDX implanted with control and EXOSC1-OE cells need to be exhibited along with the plot to enrich the in vivo experiment data.

3. As is discussed in this manuscript, EXOSC1 could potentially function as one biomarker facilitating selection of target patients who will benefit from DNA repair inhibitor therapy, supplement of retrospective clinical data analysis which manifest that KIRC patients with high level of EXOSC1 benefit more from DNA repair inhibitor therapy would be better.

*Reviewer #2 (Recommendations for the authors):*

There are some concerns on this work:

1. Figure 1E GAPDH is C>A/G>T mutation, yet CRB3 in the content was A>T/T>A, the authors should be consistent.

2. Figure 1F, have authors tried to do pathway analyses with top candidate genes to check if there are any significant pathways are associated with KIRC?

3. Figure 1I, it will have helpful to figure out among the patient mutations, how many are function related? Are there any mutations contribute to KIRC progression?

4. Please check supplemental table S3 to make sure the number of the genes are consistent with the text. There are only 30plus genes in the table, and no RAB5IF listed.

5. Figure 2E, please have a better picture, change the background color may help.

6. Figure 3, have the authors ever test the DNA cleavage activity with all the exosome components, or at least the combination of 1,7 and 8 as the authors mentioned 7 and 8 could inhibit 1 in the discussion.

7. Figure 4, to be relevant to KIRC, hot spot of VHL mutation should be used in the assay to support the author's main point of the paper.

8. Figure 4H, please be clear on groups in the figure legend, what 2,3,4,5 group mean?

9. Figure 5J, it will be nice to have survival or tumor burden data for this experiment. It will be very supportive for the manuscript.

10. Figure 6, as the authors indicated in Figure 1, four other tumor types also have high mutations, it will be interesting to examine whether DFS or OS will be different based on EXOC1 expression.

11. It will be ideal that the authors can test whether EXOSC1 has paracrine effect or not.

---

## [Author Response]

Essential revisions:1) The amount of work was sufficient to justify the conclusion. However, validation through basic experiment and clinical data analysis still requires further improvement.2) Data presentation in the figures could be adjusted more reasonable, and several questions regarding the figures remain to be explained for enriching the content and improving this work better.3) Some points in the manuscript were not in accordance with the figures, please check and make corresponding adjustments.Reviewer #1 (Recommendations for the authors):1. In line 147-148: 'KIRC displayed higher frequencies of C>A/G>T, A>T/T>A, and A>C/T>G mutations', as is manifested in Figure 1I, most frequently mutated substitutions are C>T/G>A, which is not in accordance with the description.

We are so sorry that this part was not clear in the first version. Mutation frequencies analyses indicated that the most frequently mutated substitutions in the five major cancers were C>T/G>A, but KIRC displayed higher frequencies of C>A/G>T, A>T/T>A, and A>C/T>G mutations than the other four cancers did (Figure 1I). As suggested, more information was included in the line 153-156 of the revised version.

2. In figure 6, the photographs on tumor mass from PDX implanted with control and EXOSC1-OE cells need to be exhibited along with the plot to enrich the in vivo experiment data.

As suggested, we did try to present photographs on tumors from PDX. As shown in Figure 6, the mice were executed on the different days. To take a photograph containing all the tumors from different groups, control and EXOSC1-OE tumors without Niraparib treatment were stored at -80°C. However the freeze-thaw cycle resulted in a notable shrinkage of these tumors, which might lead to misunderstanding. It is really embarrassing!

3. As is discussed in this manuscript, EXOSC1 could potentially function as one biomarker facilitating selection of target patients who will benefit from DNA repair inhibitor therapy, supplement of retrospective clinical data analysis which manifest that KIRC patients with high level of EXOSC1 benefit more from DNA repair inhibitor therapy would be better.

As suggested, we tried to retrospectively analyze the clinical data of KIRC patients treated with DNA repair inhibitor(s) to evaluate the role of EXOSC1. Due to the clinical potential of DNA repair inhibitor(s), multi clinical trials of the DNA repair inhibitor(s) such as Niraparib (NCT number: NCT03207347) in KIRC were initiated. Unfortunately, we found that all of these clinical trials are not completed or the clinical trial data are not released. We are so sorry about this. And we would be glad to respond to any further comments that you may have.

Reviewer #2 (Recommendations for the authors):There are some concerns on this work:1. Figure 1E GAPDH is C>A/G>T mutation, yet CRB3 in the content was A>T/T>A, the authors should be consistent.

As suggested, the correlation between CRB3 expression and C>A/G>T mutation was shown in the Figure 1E of the revised version.

2. Figure 1F, have authors tried to do pathway analyses with top candidate genes to check if there are any significant pathways are associated with KIRC?

Yes, we have performed pathway analyses. GO enrichment analyses indicated that, generally, these top candidate genes showed the more significant enrichment in “mitochondrial gene expression” and “organophosphate biosynthetic process” (line 130-133, Figure 1—figure supplement 1A).

3. Figure 1I, it will have helpful to figure out among the patient mutations, how many are function related? Are there any mutations contribute to KIRC progression?

As suggested, the information of function related mutations in KIRC was included in the revised version (line 156-160, Figure 1—figure supplement 1B).

4. Please check supplemental table S3 to make sure the number of the genes are consistent with the text. There are only 30plus genes in the table, and no RAB5IF listed.

We are so sorry for this terrible typo. “66” should be “36”. And C20ORF24 in Supplemental Table S3 is the previous gene name (aliase) of RAB5IF gene (Gene ID: 55969). It is really embarrassing. The corrected information was included in the line 174 and Supplemental Table S3 of the revised version.

5. Figure 2E, please have a better picture, change the background color may help.

As suggested, the background color of Figure 2E was changed in the revised version.

6. Figure 3, have the authors ever test the DNA cleavage activity with all the exosome components, or at least the combination of 1,7 and 8 as the authors mentioned 7 and 8 could inhibit 1 in the discussion.

As suggested, the DNA cleavage activity with all the exosome components (EXOSC1–9) was test. And all the exosome components (exosome complex) displayed not detect DNA cleavage activity (line 240-241, Figure 3—figure supplement 1C).

7. Figure 4, to be relevant to KIRC, hot spot of VHL mutation should be used in the assay to support the author's main point of the paper.

As suggested, a 20 nt ssDNA containing hot spot of VHL mutation (186-205) was used in the assay. And EXOSC1 cleaved the ssDNA containing hot spot of VHL mutation (line 262-263, Figure 4—figure supplement 1E).

8. Figure 4H, please be clear on groups in the figure legend, what 2,3,4,5 group mean?

We are so sorry that this part was not clear in the first version. 2, 3, 4 and 5 are the group numbers. For example, 4: the KIRC patients were grouped into 4 groups and the expression difference of EXOSC1 between the high and low mutation groups was analyzed by student’s *t*-tests. As suggested, more information was included in the figure legend of the revised version.

9. Figure 5J, it will be nice to have survival or tumor burden data for this experiment. It will be very supportive for the manuscript.

We are so sorry that this part was not clear in the first version. We supposed that Figure 6G and 6I could be used to evaluate the survival or tumor burden data of Figure 5J. And we would be glad to respond to any further comments that you may have.

10. Figure 6, as the authors indicated in Figure 1, four other tumor types also have high mutations, it will be interesting to examine whether DFS or OS will be different based on EXOC1 expression.

As suggested, we examined whether OS was different based on EXOC1 expression. Median separation KM analyses indicated that OS was not significantly different based on EXOC1 expression (BRCA *p* = 0.61, BLCA *p* = 0.20, GBM *p* = 0.72 and AML *p* = 0.17) (line 356-360, Figure 6—figure supplement 1D).

11. It will be ideal that the authors can test whether EXOSC1 has paracrine effect or not.

As suggested, overexpression of EXOSC1 in MEF increased the production of interferon b (IFNb). However, overexpression of EXOSC1 showed little impact on the production of IFNb in cancer cells, which might be because that cGAS/sting pathway is frequently inactivated in the most cancer cells.